# Identifying and assessing nonlinear drivers of capelin and Atlantic cod population dynamics using empirical dynamic modelling (EDM) scenario exploration

Reid W. Steele[1,2]*, Jin Gao[1,3], Mariano Koen-Alonso[4], Paul M. Regular[4]

**1** Center for Fisheries Ecosystem Research (CFER), Fisheries and Marine Institute of Memorial University of Newfoundland, Canada, **2** Department of Biology, Dalhousie University, Halifax, Canada, **3** Fraser Sockeye and Pink Salmon Analytical Program, Fisheries and Oceans Canada, Delta, Canada, **4** Northwest Atlantic Fisheries Centre (NAFC), Fisheries and Oceans Canada, Canada

* Reid.Steele2@gmail.com

## Abstract

As anthropogenic stressors such as climate change and fisheries continue to disrupt marine ecosystems, it is becoming more important to understand and predict resulting nonlinear effects on species population dynamics. The marine ecosystem of the Newfoundland Shelf serves as a unique case study for this purpose. Capelin (*Mallotus villosus*) and Atlantic cod (*Gadus morhua*) were dominant species of the Newfoundland Shelf community before populations of both species collapsed in the early 1990s during a period of abnormally cold climate and overfishing of groundfish, culminating in a regime shift. In this study, we examine 12 climatic and ecological covariates for driving effects on capelin and cod population dynamics using Convergent Cross-Mapping (CCM). We then examine the magnitude and direction of identified drivers on capelin and cod population dynamics, and identify nonlinear relationships between these drivers and resulting population dynamics using Empirical Dynamic Modelling (EDM) scenario exploration. The drivers of cod and capelin were mostly common and synergistic. For both cod and capelin, we found long-term climatic change was the strongest climatic driver of biomass, with warming temperatures predicted to increase biomass for both species in the near future. We observed nonlinearity in this relationship for capelin, with warming temperatures becoming a negative driver at high biomass. Capelin and cod biomass were the strongest ecological drivers of each other, with increasing biomass in either benefitting the other. Overall, this study shows EDM scenario exploration can be used to identify primary drivers of a species' population dynamics, and predict nonlinear reactions to environmental and/or ecological changes in their environment.

**Data availability statement:** North Atlantic Fisheries Organization catch data for cod and capelin is available from the STATLANT 21A database https://www.nafo.int/Data/STATLANT-21A. Newfoundland and Labrador Climate Index Data is available from https://doi.org/10.20383/101.0301 . Cod and capelin biomass index data and ice timing data are available from https://doi.org/10.5281/zenodo.17515115 . Code is available from https://github.com/reidsteele2/Capelin-Cod-Scenario-Exploration.

**Funding:** J. G. RGPIN-2021-03249 National Science and Engineering Research Council of Canada (NSERC) https://www.nserc-crsng.gc.ca/index_eng.asp Funders played no role in study design, data collection and analysis, decision to publish, or manuscript preparation.

**Competing interests:** The authors have declared that no competing interests exist.

## Introduction

As marine ecosystems continue to change due to climate change, fishing, and other stressors, it is becoming increasingly necessary to understand and predict how species population dynamics will react to these changes [1,2]. Systems that have a history of significant change offer an avenue for identifying and contrasting the causes and effects of such changes. The marine ecosystem of the Newfoundland Shelf may be a particularly useful study system given the major fisheries and ecosystem induced changes that have been observed in the region.

Capelin (*Mallotus villosus*) and Atlantic cod (*Gadus morhua*) were dominant species of the Newfoundland Shelf community before both species collapsed in the early 1990s during a period of abnormally cold climate and overfishing of groundfish [3–6]. Following this collapse, both capelin and cod mature earlier [3,7], and capelin spawn later in the year [8]. With the exception of a brief partial recovery through the mid 2010s, both capelin and cod continue to show low productivity and altered life history strategies despite reduced fishing pressure and a return to average climate conditions [3,8–11].

Previous study on capelin population dynamics reveals ice timing, capelin condition, larval capelin abundance, and differences in the pre- and post-collapse states as potential drivers of capelin biomass and spawn timing [9,12]. However, separate consideration of pre- and post-collapse capelin dynamics may miss common drivers between both periods, which may be difficult to impossible to identify using linear modelling due to the nonlinear, biphasic nature of capelin biomass time series, and capelin condition and larval capelin abundance are downstream effects of potentially unknown upstream drivers. Similarly, literature suggests capelin imposes strong bottom-up control on cod, and cod recovery may require a previous recovery of capelin [13,14], necessitating understanding of upstream drivers of capelin population dynamics to predict those of cod. Predicting what conditions may lead to cod and capelin recovery or further decline is necessary to responsibly manage these species into the future.

One potential tool for this purpose is Empirical Dynamic Modelling (EDM) scenario exploration [15], as this method can be used to predict how species react to changes in climatic conditions, fishing pressure, or biomass of other species in the ecosystem [16]. As an example, scenario exploration has been used to predict how changes in temperature affect sardine abundance, and how this relationship relates to fishing pressure, and management of future sardine biomass [17]. EDM is capable of identifying causative covariates which drive time series of fish biomass using Convergent Cross-Mapping (CCM) [18], and using these drivers to generate nonlinear predictions [17,19–23], including for the stocks examined in this study [24,25]. However, these models do not directly show the magnitude and direction causative drivers have on fish biomass, or how they may change with changing ecosystem conditions. Scenario exploration seeks to solve this issue by comparing how EDM model predictions change when the values of driving covariates are perturbed.

In this study, we aim to identify drivers of capelin and cod biomass dynamics using CCM, and use scenario exploration to assess the magnitude and direction of each driver's effect on capelin and cod biomass and how the magnitude and direction change with the relative value of the driver and/or biomass.

## Results

Univariate Convergent Cross-Mapping (CCM) testing results are in Table 1. CCM testing returned significant results for catch, Greenland halibut biomass, the Cumulative Newfoundland and Labrador Climate Index (CNLCI), and Sea Surface Temperature (SST) across both capelin and cod, identifying them as potential drivers of capelin and cod biomass. Ice timing and air temperature also returned significant CCM results for capelin. No covariates returned significant results for cod and not for capelin. Cod and capelin also returned significant cross-map results against each other, indicating strong synchrony between the two species.

To identify the magnitude and direction of the effects of potential drivers on cod and capelin biomass, we performed scenario exploration on cod and capelin time series using every driver identified by CCM testing. To validate the models used for scenario exploration, we calculated leave-one-out cross validation prediction skill ($\rho$) of S-Map models for all drivers identified by CCM, which are reported in Table 1. All models achieved $\rho$ values of 0.6 or higher with the exception of ice timing predicting capelin biomass and capelin predicting cod biomass. The latter value was heavily biased by very poor predictions in the pre-collapse period, and improved to a $\rho$ value of 0.793 when excluding 1987–1991. We then calculated the effect perturbing the values of each driver had on predicted capelin and cod biomass ($\Delta B/\Delta X$). Positive $\Delta B/\Delta X$ values indicate the driver had a positive effect on predicted biomass, while negative $\Delta B/\Delta X$ values indicate the opposite.

Univariate scenario exploration revealed SST, CLNCI, Greenland halibut biomass, and catch had primarily positive effects on both capelin and cod (Fig 1). Capelin and cod also had primarily positive effects on each other. Air temperature and ice timing effects were slightly negative overall but were generally weak and outlier driven. For both cod and capelin, catch and CLNCI had the largest effects on $\Delta B/\Delta X$, followed by their effects on each other. The positive effects of SST and Greenland halibut on $\Delta B/\Delta X$ were weaker and more outlier-driven (Fig 1).

In univariate scenario explorations using climatic drivers, capelin was predicted to benefit from warming climate during the collapsed period and stable climate in the pre-collapse period. In the post-collapse period, $\Delta B/\Delta SST$ is near zero at average and low SST (<1), and begins to increase linearly at high SST (>1, Fig 2). This indicates the model predicts capelin biomass increasing with warming temperatures in the post-collapse period, and average to low temperatures predict no change in biomass. The same pattern is visible in $\Delta B/\Delta CNLCI$ (Fig 2). In the pre-collapse period, $\Delta B/\Delta SST$ is positive

**Table 1. p-values for Convergent Cross-Mapping (CCM) significance testing against all covariates for both capelin and Atlantic cod (\* indicates significant result, p<0.05) and leave-one-out cross-validation prediction skill ($\rho$) of multivariate S-Map models using potential drivers of capelin and cod biomass as identified by CCM.**

| Covariate | Source/Description | CCM p-value | | Cross-validation $\rho$ | |
|---|---|---|---|---|---|
| | | Capelin | Cod | Capelin | Cod |
| Capelin | DFO, Mowbray (2012) | | **0.039\*** | | −0.032 |
| Atlantic Cod | DFO, Doubleday (1981) | **<0.001\*** | | 0.631 | |
| Catch | NAFO (2021) | **<0.001\*** | **<0.001\*** | 0.677 | 0.858 |
| Greenland Halibut | DFO, Doubleday (1981) | **0.015\*** | **0.008\*** | 0.828 | 0.901 |
| Cumulative NLCI | Cyr & Galbraith (2021) | **<0.001\*** | **<0.001\*** | 0.677 | 0.858 |
| Annual NLCI | Cyr & Galbraith (2021) | 0.220 | 0.107 | | |
| Winter NAO | Cyr & Galbraith (2021) | 0.163 | 0.342 | | |
| Sea Ice | Cyr & Galbraith (2021) | 0.594 | 0.162 | | |
| Ice Timing | Cyr & Galbraith (2021) | **0.009\*** | 0.673 | 0.406 | |
| Air Temperature | Cyr & Galbraith (2021) | **0.026\*** | 0.060 | 0.604 | |
| SST | Cyr & Galbraith (2021) | **0.001\*** | **0.028\*** | 0.617 | 0.883 |
| Bottom Temperature | Mowbray (2012) | 0.149 | 0.089 | | |

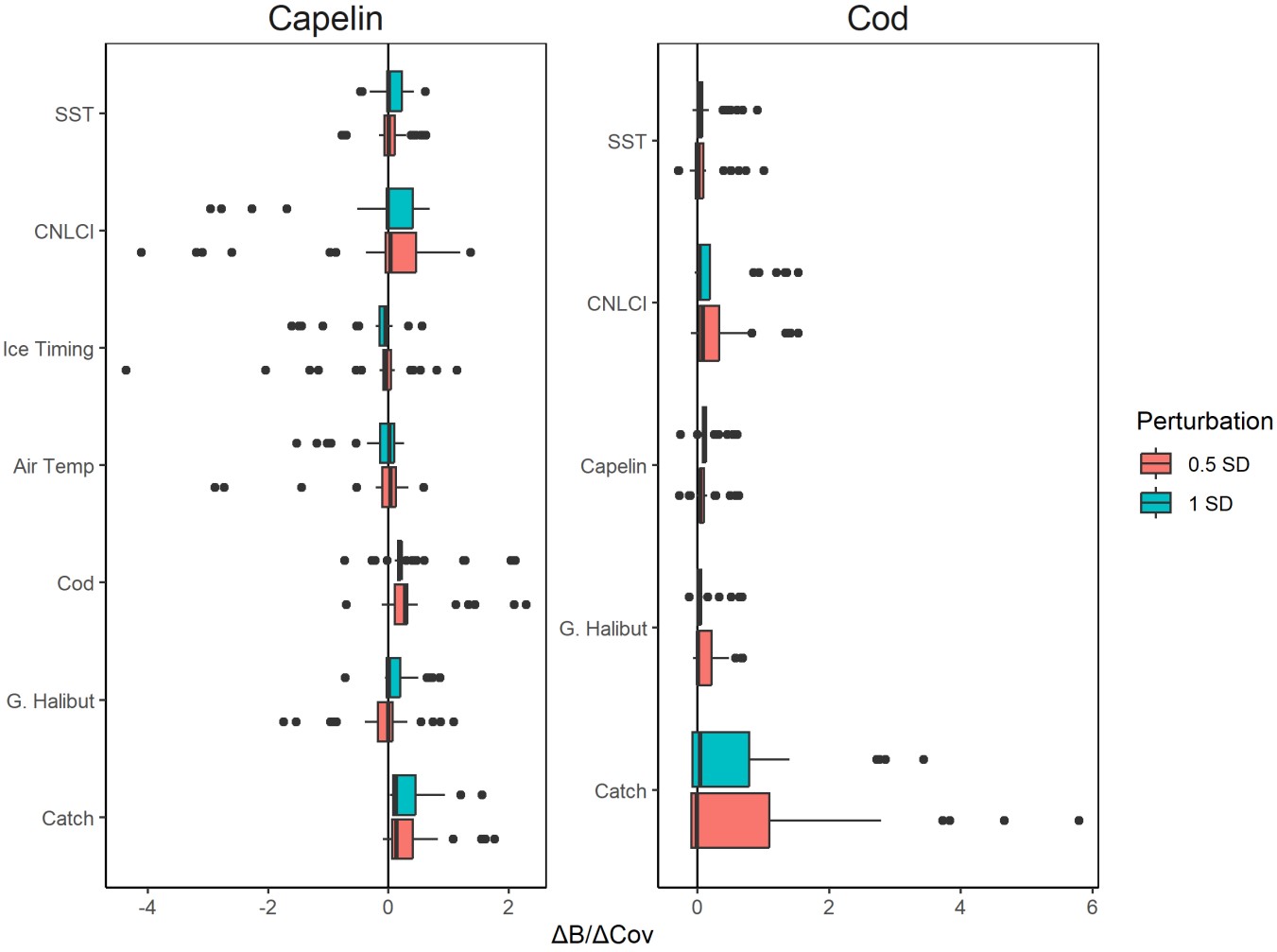

**Fig 1. Boxplots of predicted change in normalized capelin acoustic index (left) and cod bottom trawl index (right) per change in driver (ΔB/ΔX) calculated from EDM scenario exploration using ±0.5 standard deviation (red) and ±1 standard deviation (green) perturbation scenarios.**

at low SST and negative at high SST (Fig 2), indicating warming temperatures would benefit capelin biomass when SST is low, and vice versa. ΔB/ΔCNLCI is negative through the full pre-collapse period, indicating warming climate would predict capelin biomass decline pre-collapse. This would agree with the ΔB/ΔSST interpretation that stable climate is predicted to benefit capelin biomass in the pre-collapse period, though there are no low ΔB/ΔCNLCI values in the pre-collapse period to confirm this. (Fig 2). Air temperature and ice timing similarly negatively affected ΔB/ΔX when increased during the pre-collapse period, though ΔB/ΔX was slightly increased by increased air temperature and decreased by increased ice timing for most of the post-collapse period (Fig 2).

Counterintuitively, capelin biomass was predicted to increase from increased presence of predators and increased catch. Through the entire time series, capelin predictions and ΔB/ΔX consistently increased with increased catch and increased cod biomass (Fig 3). Increased Greenland halibut biomass increased capelin biomass predictions except for the pre-collapse period using the 0.5 standard deviation perturbation, for which this trend was flipped (Fig 3). ΔB/ΔGH trended upwards with Greenland halibut biomass at high values, though the relative lack of years contributing to this pattern suggests it could be due to chance (Fig 3).

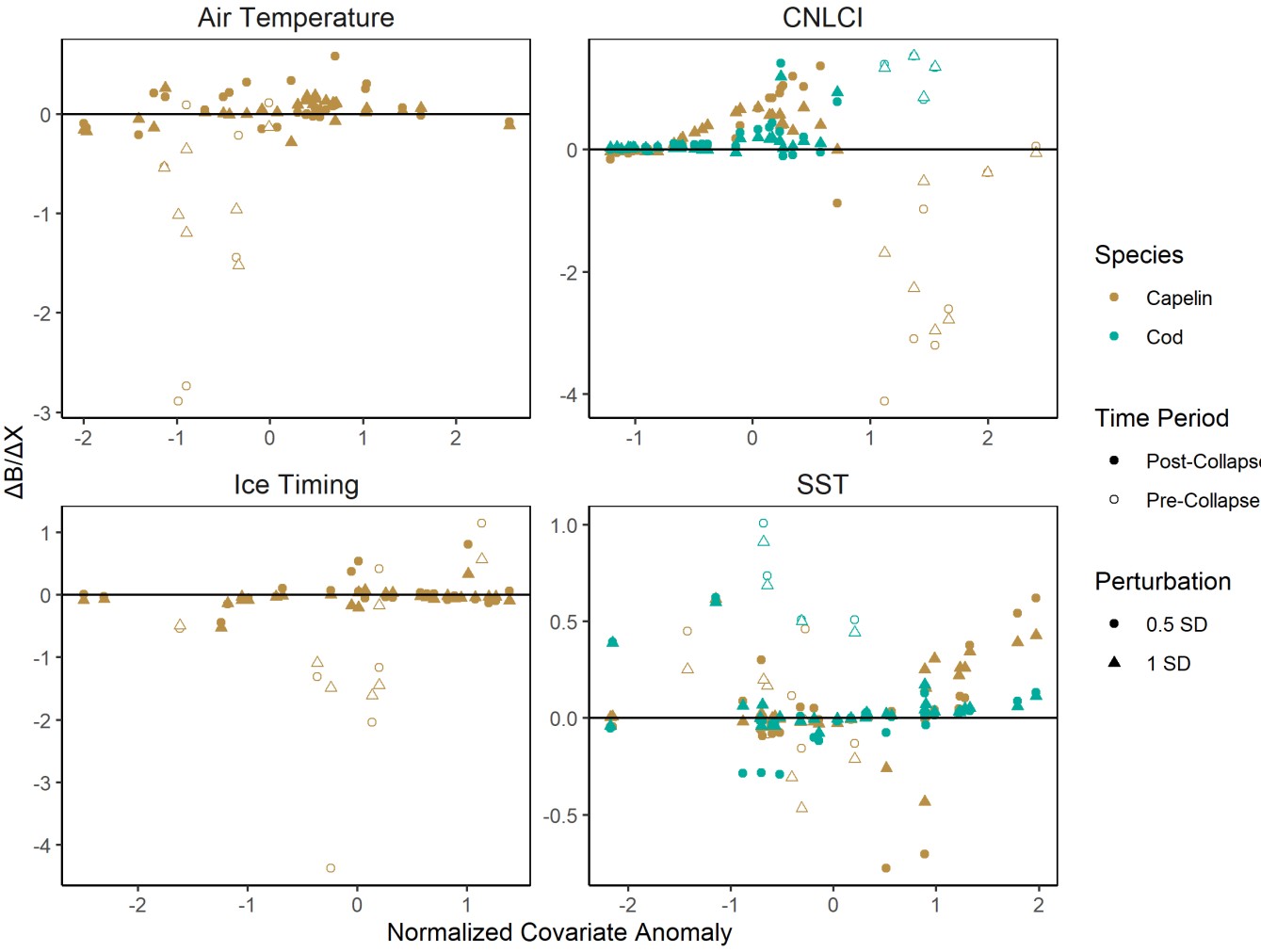

**Fig 2. Scatterplots of the difference between positive perturbation predictions and negative perturbation predictions for each year in the capelin acoustic index (brown) and cod (green) bottom trawl index using S-Map scenario exploration with climate drivers perturbed positively and negatively by a half standard deviation (circle) and a full standard deviation (triangle) from 1984-2019, distinguished between years before (filled) and after (open) the 1991 capelin collapse.** Relationships for covariates returning statistically insignificant convergent cross-mapping results are excluded.

Across all scenarios, Atlantic cod biomass predictions consistently increased with both climate and ecological drivers. Cod biomass predictions increased with SST in the pre-collapse period and slightly during the post-collapse period and showed a clear increasing pattern of $\Delta B/\Delta SST$ with SST over both periods (Fig 2). This pattern was repeated in CLNCI (Fig 3). Increased catch perturbations often resulted in extreme, unrealistic increases in cod biomass predictions. (Fig 2). Perturbing capelin biomass resulted in cod biomass predictions increasing with capelin in every year of the time series, though this increase was much larger in the post-collapse period (Fig 3). There was no clear pattern between capelin biomass and cod $\Delta B/\Delta Capelin$ (Fig 3). Lastly, increasing Greenland halibut biomass similarly greatly increased cod predictions in the pre-collapse period and slightly increased them in most of the post-collapse period, but conversely exhibited a decreasing pattern in $\Delta B/\Delta GH$ with increasing Greenland halibut biomass (Fig 3).

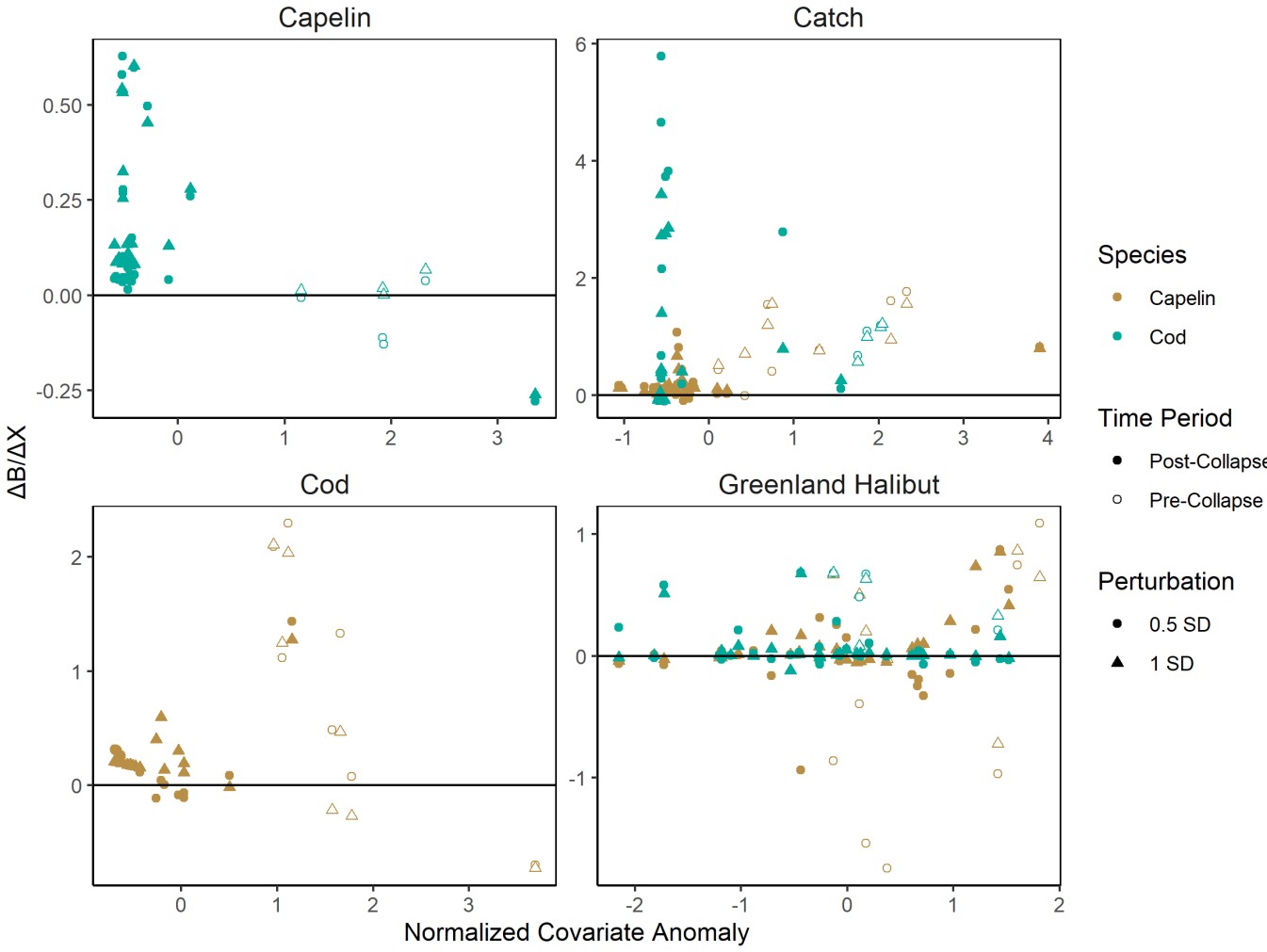

**Fig 3. Scatterplots of the difference between positive perturbation predictions and negative perturbation predictions for each year in the capelin acoustic index (brown) and cod bottom trawl index (green) using S-Map scenario exploration with ecological drivers perturbed positively and negatively by a half standard deviation (circle) and a full standard deviation (triangle) from 1984-2019, distinguished between years before (open) and after (filled) the 1991 capelin collapse.** Relationships for covariates returning statistically insignificant convergent cross-mapping results are excluded.

One limitation of univariate scenario exploration is that synergistic or antagonistic relationships may be overlooked. Although multivariate scenario exploration introduces its own complexities (e.g., drivers must be perturbed in ratios that are realistic to each other), it may reveal relationships missed using the univariate approach. We therefore performed multivariate scenario exploration on cod using the common climatic drivers of cod and capelin biomass (CNLCI and SST) as in the univariate example, and added the predicted capelin biomass given those perturbations as a second driver. Multivariate scenario exploration results largely mirrored the univariate results given the same climatic drivers (Fig 4, Fig 2), indicating the climatic drivers had a greater impact on predictions than the resulting capelin predictions. No strongly synergistic or antagonistic interactions were detected.

## Discussion

In this study, EDM scenario exploration was able to identify and rank drivers of both capelin and cod biomass, illuminating clear relationships between biomass drivers and the direction and degree to which they drive biomass. The drivers of

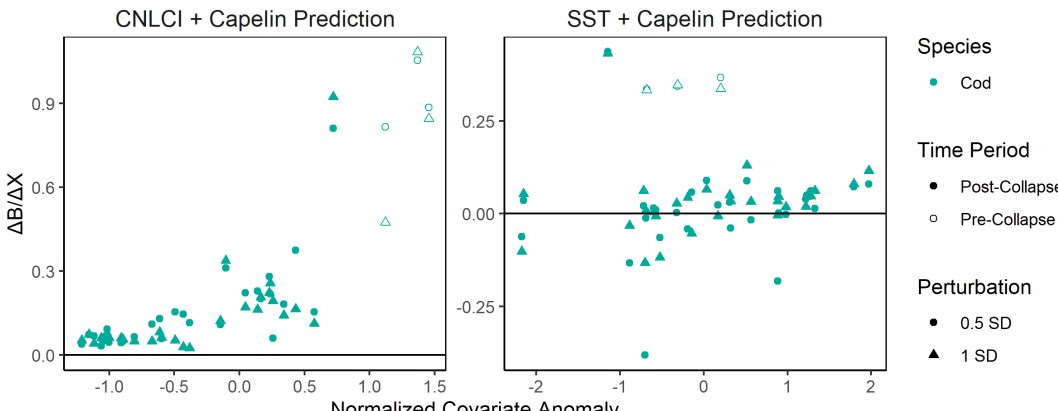

**Fig 4. Scatterplots of the difference between positive perturbation predictions and negative perturbation predictions for each year in the cod bottom trawl index from 1984-2019 using multivariate S-Map scenario exploration with common climate drivers of capelin and cod perturbed positively and negatively by a half standard deviation (circle) and a full standard deviation (triangle), and the resulting capelin biomass prediction used as drivers, distinguished between years before (open) and after (filled) the 1991 capelin collapse.**

cod and capelin were mostly common and synergistic. For both cod and capelin, we found long-term climatic change as measured by the CLNCI was the strongest climatic driver of biomass. Ecologically, scenario exploration indicated both species' strongest drivers were catch and each other. However, both analyses suffer from synchrony, meaning the relationship between the two variables was so tight that the model could not discern causal direction [18]. All of these effects are positive (Fig 3), indicating that the dominant processes are bottom-up. Apparent effects of cod on capelin and catch on both species are spurious correlations born of strong feedback between target species biomass and its associated driver. In contrast, the driving effect of capelin on cod appears to represent a true causal relationship.

Our finding that capelin availability drives cod biomass matches previous literature surrounding cod population dynamics. Availability of capelin has been found to improve cod growth, condition, and gonad size in Northern cod [14], and cod starvation due to the collapse of capelin has been identified as an important contributor to the natural mortality and slow recovery of Northern cod [13]. Capelin availability also exhibits similar driving effects on cod population dynamics in the Barents Sea, despite vastly different population levels and trends in both species as compared to the Newfoundland Shelf [26]. This also provides a basis for our finding that capelin and cod biomass were driven synergistically by common drivers, such as SST and CNLCI. Conversely, the reverse effect of cod driving capelin is not as well supported. Cod predation is included in capelin stock assessment models in the Barents Sea only to ensure sufficient escapement of mature capelin for adequate recruitment [27], and Newfoundland Shelf capelin (and the ecosystem as a whole) are widely considered to be driven by bottom-up processes [9,28]. This is consistent with the synchronous positive result we found examining the driving effect of capelin on cod – top-down pressure from cod predating on capelin would appear as a negative relationship between cod and capelin biomass, which is the opposite of our finding. Most likely, increases in cod biomass indicate that conditions are also favourable for capelin, and the system is largely bottom-up driven. A similar argument could also be made to explain the positive effect of Greenland halibut on capelin biomass (Fig 3).

We also found that warming climate was generally beneficial for both capelin and cod across all climatic drivers, though the magnitude of this benefit, and the conditions in which it manifested differed. For example, warming climate as measured by both the CNLCI and post-collapse SST had a negligible effect on either cod or capelin until a certain threshold was reached, and capelin exhibited a second threshold after which warming climate began to have a negative effect on biomass (Fig 2). This agrees with previous research finding warmer temperatures are associated with increased biomass in the finfish community throughout the last 70 years [29]. Warming temperatures are also associated with earlier

spawning and stronger recruitment for this capelin stock [8]. Since capelin collapsed, they have spawned about three weeks later than they did pre-collapse, leading to a mismatch between larval capelin and their zooplankton prey [30], which may be alleviated by increased growth resulting from higher temperatures [31]. If this holds true, recent warming conditions throughout the early 2020s may herald some recovery of capelin, and thus cod, some signs of which are already visible through recent increases in larval capelin abundance and capelin condition [32]. However, this mechanism is speculative, and future research is required to confirm both the effects of increasing temperatures on Northern cod and capelin, and the mechanism(s) which are driving it [29].

Multivariate scenario explorations of cod biomass with climate drivers and predicted capelin biomass returned similar results to univariate scenario explorations of the same climate drivers (Fig 4, Fig 2), indicating that either climatic drivers were more impactful on cod biomass than the resulting capelin predictions, or the effect of climate on cod biomass via capelin biomass is already largely accounted for in climate drivers alone. The wealth of research discussed above on the impact of capelin availability on cod [13,14,26] would suggest the latter is more likely. CNLCI is a particularly strong driver of capelin biomass [25], which may overpower the effect of capelin biomass on cod biomass due to its collinearity with the former. Still, multivariate scenario exploration provides an interesting avenue to explore the relative relationships between drivers and their effects on the target time series. Further research might explore how to realistically perturb drivers in relation to each other to extend such analyses to more driver combinations.

The main weakness of EDM scenario exploration in this study is its inability to discern directionality in closely related time series, potentially leading to false positive identification of drivers, and unrealistic predictions. This weakness is most obvious in the positive relationship returned between catch and biomass for both capelin and cod. Logically, it would be expected that catch would lead to decreased biomass, especially for the 2J3KL Atlantic cod stock, which has a well-documented history of being overfished [4–6]. The positive relationship returned by scenario exploration is likely a result of catch acting as an index of cod biomass, which is misinterpreted as causality due to synchrony [18]. This also may be partially explained by the years covered by the dataset, which includes many years after this stock was placed under moratorium in 1992, and does not include many years of high fishing pressure in the 1960s and 1970s before biomass collapsed [10]. It is possible that a different relationship could be detected using other metrics which better account for the proportion of fishing mortality being experienced by the stock, such as fishing effort or estimates of fishing mortality, or if a longer time series were used [19]. However, the time series we use are generally considered the most accurate biomass indices available for 2J3KL capelin and cod, potential longer fisheries-dependent time series are vulnerable to bias, and fishing effort data reporting for this stock is not generally considered to be reliable. Regardless, this still serves as an avenue for potential future research into the utility of EDM scenario exploration in fisheries ecology.

Another potential weakness of scenario exploration in this study is our focus on single driver models. In reality, many environmental and ecological drivers may be constantly changing in response to each other, which complicates predictions in comparison to the single driver tests we perform in this study. Multivariate EDM models can be used to model capelin population dynamics effectively [25], and focusing on single drivers may result in missed synergistic and/or antagonistic interactions between drivers. Perturbing the values of multiple drivers at once in a realistic way is challenging, as the values of drivers are unlikely to relate to each other directly and the benefit of including multiple drivers may be diminished by collinearity. We provide one example of how the relationship between drivers might be used to perform multivariate scenario exploration, but this example is limited, and only possible because of the clear, well-studied forcing relationships between climate, capelin, and cod. There are also other potential issues with multivariate scenario exploration: For example, increasing the number of drivers exponentially increases the number of possible driver combinations to model, which may cause computational issues. Testing a large number of drivers or driver combination also increases the risk of false positives. Irrelevant covariates may be filtered out using CCM significance testing, but this process is not infallible, and potentially false drivers may slip through the cracks, in this study including synchronous results such as catch, or driving effects that are weak and possibly spurious, such as Greenland halibut. What covariates (and how many) are used

in CCM and scenario exploration should be carefully considered, and results should be logically scrutinized and compared to literature to account for such possibilities. Still, the potential for interactions between driving effects may make further research into multivariate scenario exploration worthwhile.

Scenario exploration is also unable to account for missing information, such as target and/or driver values which do not occur in our dataset, or potential drivers which we did not include in our analyses. As an example, it remains to be seen whether a return to the pre-collapse condition in long-term climate state would result in recovery for capelin and cod, or if such a recovery would be hampered by other mechanisms, such as their altered life history strategies since the collapse. As these time series grow and experience new phases, these interactions, and the ability of EDM to account for them, should be clarified. Though careful thought would be required to perturb multiple drivers realistically, multivariate scenario exploration remains an interesting avenue for potential research into the use of EDM scenario exploration as a tool for stock assessment.

Regardless, this study shows EDM scenario exploration can be utilized to scope how species may react to environmental and/or ecological changes in their environment. This information can be applied in the context of management to explore how species would be expected to respond to changes in the environment, management actions, or a combination of both [17]. For example, EDM scenario exploration could be combined with climate change projections [33], fishing scenarios [19], or some combination of both to predict how species may react to different stressors, or combinations of stressors. In the context of cod and capelin, future research into scenario exploration comprising more detailed catch information, more species, and more specific potential future scenarios would be beneficial to fully explore the potential of scenario exploration as a tool for their management.

More broadly, as ecosystem approaches become more common in the management of fisheries, and with them the consideration of trade-offs among fisheries, the increasing impacts of climate change, and the consideration of strategic horizons (i.e., medium-long term planning) in addition to the typical tactical horizons (i.e., short-term planning) of fisheries management decisions, EDM scenario exploration can provide a useful platform for scoping the interactions and drivers that may deserve closer attention and deeper examination. EDM contains a complex suite of tools, but they are easier to implement and explore than many stock-assessment and ecosystem models, allowing them to act as an effective first pass strategy for a data-driven identification of the relevant nonlinear interactions and drivers that more sophisticated and onerous models may need to consider.

## Methods

### Data series

Data series and their sources are listed in Table 1. Capelin biomass dynamics were gathered from the Fisheries and Oceans Canada (DFO) spring acoustic survey [34]. 9 missing years in the time series were filled by interpolation using Gaussian Process regression fit using maximum likelihood estimation (See Supplementary S1). To prevent regression towards pre-collapse values after the capelin collapse, post-collapse values were interpolated using only post-collapse data. Cod biomass dynamics were derived from the DFO fall random stratified bottom trawl survey index in NAFO divisions 2J3KL [35,36]. The Newfoundland Climate Index (NLCI) [11], was used as a general indicator of climatic conditions. Likewise, the cumulative NLCI (CNLCI) was used to capture cumulative effects of warm or cold climate across multiple years. To test the influence of individual climatic factors, ecologically component parts of the NLCI were also tested as individual covariates. Winter North Atlantic Oscillation (NAO) serves as a high level climatic driver, associated with temperature and ice conditions [37]. Sea Surface Temperature (SST) is a commonly used environmental covariate, which has been shown to drive forage fish populations in other systems [17,38]. Air temperature is a strong proxy of SST [39]. Bottom temperature was also tested, as capelin and cod are both primarily demersal species. Day of year of sea ice retreat (henceforth referred to as ice timing) was used as an indicator of the timing of spring bloom and its effects on

upper trophic levels [9]. Finally, Greenland halibut biomass and fisheries catch of capelin and cod were also included to account for potential top-down and/or competition effects. Greenland halibut was chosen for this purpose as it is, together with cod, one of the most important fish top predators in this system.

## EDM analyses

Univariate simplex projections were used to calculate the optimal embedding dimension ($E$) in the range of 1–5, and univariate S-Map projections were used to calculate the optimal weighting parameter ($\theta$) in the range of 0.01 to 9 for both the capelin acoustic index and cod bottom trawl survey index. To narrow the range of covariates considered for scenario exploration, capelin and cod biomass indices at optimal $E$ were tested for causative relationships against all covariates using convergent cross mapping (CCM). Significance of CCM results was tested by producing p-values by comparison to a null distribution of 1000 CCMs [40] using phase-randomized surrogate datasets [41]. To validate the models used for scenario exploration, leave-one-out cross validation of S-Map models using each driver was performed, and prediction skill ($\rho$) was calculated for all drivers identified by CCM. For cross-validation tests and scenario exploration, $E$ was set to 2 for capelin to accommodate multivariate modelling ($E=1$ was optimal, $\Delta\rho=0.034$), and $E$ was set to the optimal value of 4 for cod.

## Scenario exploration

Scenario exploration methodology was adapted from previous literature [15]. Data were time-delay embedded (*i.e.,* separated into lags) and converted to normalized anomalies before scenario exploration. For both cod and capelin, multivariate S-Maps predicting the biomass index ($B$) one year in the future ($t+1$) were fit using the previously calculated optimal weighting parameter $\theta$ and $E$ lags of the biomass index and the most recent year of each covariate ($X$) which returned significant CCM results (henceforth referred to as drivers):

$$B(t+1) = f[B(t),\ B(t-1),\ldots(B(t-(E-1)),\ X(t)],$$

These S-Maps were then used to run predictions on the same dataset with the drivers perturbed positively and negatively by a set value ($\Delta$):

$$B_+(t+1) = f[B(t),\ B(t-1),\ldots B(t-(E-1)),\ X(t)+\Delta], \tag{1}$$

$$B_-(t+1) = f[B(t),\ B(t-1),\ldots B(t-(E-1)),\ X(t)-\Delta], \tag{2}$$

$\Delta$ values of 0.5 and 1 (corresponding to half a standard deviation and one standard deviation) were used, yielding four perturbation scenarios. Time series of all scenarios are plotted in Supplementary S1. The magnitude and direction of each driver's influence on the biomass indices per change in the driver (henceforth referred to as $\Delta B/\Delta X$) were calculated by subtracting the positive perturbation predictions by the negative perturbation predictions, and dividing by the total perturbed difference in the driver to get a rate, for each year in the time series:

$$\frac{\Delta B}{\Delta X} = \frac{B_+(t+1)-B_-(t+1)}{[X(t)\ +\ \Delta]\ -\ [X(t)\ -\ \Delta]}, \tag{3}$$

where *[X(t) + Δ] − [X(t) − Δ] = 2Δ*. Boxplots of $\Delta B/\Delta X$ for all driver and $\Delta$ combinations were compared to assess the relative magnitude and overall direction of the influence of each driver on capelin and cod biomass indices. For each driver and $\Delta$ combination, plots of $\Delta B/\Delta X$ against the driver were produced to assess the relationship between the value of the driver and its resulting influence on biomass, and how this influence changed over time and with the value of the driver.

## Multivariate scenario exploration

In some cases, drivers may act synergistically or antagonistically, necessitating the use of multivariate models to account for the cumulated effects of multiple drivers. However, this process is complicated by the relationships between drivers, meaning perturbing multiple drivers by one standard deviation as in the univariate case may not present a realistic scenario. To provide a realistic example, we performed multivariate scenario exploration of cod biomass using the common climatic drivers of cod and capelin biomass (CNLCI and SST) and their univariate scenario exploitation capelin biomass predictions as the drivers, ensuring the perturbations of the climatic drivers and capelin biomass can coexist. These multivariate scenario explorations are expressed as:

$$B_{cod+}(t+1) = f[B(t),\ B(t-1), \ldots B(t-(E-1)),\ X(t) + \Delta, B_{capelin+}(t)], \tag{4}$$

$$B_{cod-}(t+1) = f[B(t),\ B(t-1), \ldots B(t-(E-1)),\ X(t) - \Delta], B_{capelin-}(t)] \tag{5}$$

where $B_{capelin+}$ and $B_{capelin-}$ are calculated according to equations 1 and 2 using the same perturbed driver $X(t)$. $\Delta B/\Delta X$ was then calculated according to equation 5.

## Supporting information

**S1 File. Raw data and individual scenario exploration figures.**
(DOCX)

## Author contributions

**Conceptualization:** Reid W Steele, Jin Gao.

**Data curation:** Reid W Steele, Mariano Koen-Alonso, Paul M. Regular.

**Formal analysis:** Reid W Steele.

**Funding acquisition:** Jin Gao.

**Investigation:** Reid W Steele, Jin Gao.

**Methodology:** Reid W Steele, Jin Gao.

**Project administration:** Jin Gao.

**Resources:** Jin Gao, Mariano Koen-Alonso.

**Software:** Reid W Steele.

**Supervision:** Jin Gao, Mariano Koen-Alonso, Paul M. Regular.

**Validation:** Reid W Steele, Jin Gao, Mariano Koen-Alonso, Paul M. Regular.

**Visualization:** Reid W Steele, Paul M. Regular.

**Writing – original draft:** Reid W Steele.

**Writing – review & editing:** Reid W Steele, Jin Gao, Mariano Koen-Alonso, Paul M. Regular.

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
