## [Decision Letter · Decision Letter 0]

13 Sep 2025

Dear Dr. Steele,

Thank you for submitting your manuscript to PLOS ONE. After careful consideration, we feel that it has merit but does not fully meet PLOS ONE’s publication criteria as it currently stands. Therefore, we invite you to submit a revised version of the manuscript that addresses the points raised during the review process.

We look forward to receiving your revised manuscript.

Kind regards,

Geir Ottersen

Academic Editor

PLOS ONE

Journal Requirements:

3. In the online submission form, you indicated that ice timing, capelin acoustic index data and trawl survey data cannot be archived publicly because they are collected by and property of Fisheries and Oceans Canada. However, they will be made available to individuals upon direct request to the authors.

Newfoundland and Labrador Climate Index data is available at https://doi.org/10.20383/101.0301.

North Atlantic Fisheries Organization catch data for cod and capelin is available from the STATLANT 21A database https://www.nafo.int/Data/STATLANT-21A .

Reviewers' comments:

Reviewer's Responses to Questions

**Comments to the Author**

1. Is the manuscript technically sound, and do the data support the conclusions?

Reviewer #1: Partly

Reviewer #2: Yes

2. Has the statistical analysis been performed appropriately and rigorously?

Reviewer #1: No

Reviewer #2: Yes

3. Have the authors made all data underlying the findings in their manuscript fully available?

Reviewer #1: No

Reviewer #2: Yes

4. Is the manuscript presented in an intelligible fashion and written in standard English?

Reviewer #1: Yes

Reviewer #2: Yes

Reviewer #1: Review of PONE-D-25-37653

“Identifying and assessing nonlinear drivers of capelin and Atlantic cod population

dynamics using Empirical Dynamic Modelling (EDM) scenario exploration” by Steele et al.

This paper uses an EDM fitted to time-series for capelin and cod of Newfoundland to try to identify drivers, potentially non-linear, on population dynamics.

My main concern with this manuscript is the short time series (many missing years for capelin, 29 years) and the large number of drivers tested (11) and potential overparameterization issues. Also, in ref. 15, it is recommended that interpolations should only be done to small changes in smoothly varying data, which seems to be violated (Fig. S1, at least in the 1980s and maybe also after collapse?). How much it the extensive data imputation (9 data points vs. 29 real data) affect the results? Unfortunately, it is hard to understand what exactly was done, as the data is not described and presented in enough detail (Table 1 is not enough). For example, cod time-series, the main investigated species together with capelin, is not presented! Also, the different drivers were included in the analysis is not well motivated. For example, air temperature is a proxy for SST, which is justified with a paper based on lakes (38). SST may not be the most relevant proxy for temperatures experienced by cod, which is often associated with deeper water. But then all three (SST, Air Temperature and bottom temperature) are included in the analysis. The same goes for the climate indices; why are three different ones used? Greenland halibut is included as competitor/predator, but with no reference! Could other species have been included instead (Haddock, Pollock, Dogfish, American Plaice, other)? Please justify! To me, it seems strange and unbalanced to use three temperature proxies + three climate proxies and just one competitor/predator. The highly unexpected results of capelin benefiting from both harvesting and predators may well be a result of a poorly specified starting model. That the system is bottom up and caplin drives cod dynamics (lines 160- 164) is highly speculative, especially if it simultaneously applies to long-lived Greenland halibut. This needs to be either flagged as speculation or referenced more carefully. Furthermore, I am surprised that a paper on capelin collapse dynamics do not even mention the work of Frank et al. 2016 (https://doi.org/10.3354/meps11761 and other work from this group), and the possibility that there has been no collapse of capelin, just a mismatch in the survey and the distribution of capelin. They found no indication of effects of capelin collapse on cod dynamics, which seems to be in direct opposition to the claims on lines 160 -164? Please clarify!

Despite these critical remarks, if the authors can give more explanation and justification of the model and data, I believe that the general approach of the paper is scientifically sound, and the scenario investigation may be of interest.

Reviewer #2: See attached Review Comments.

**Do you want your identity to be public for this peer review?** For information about this choice, including consent withdrawal, please see our Privacy Policy

Reviewer #1: No

Reviewer #2: No

---

## [Author Response · Author response to Decision Letter 1]

10 Nov 2025

Reviewer and editor comments are addressed directly in the Response to Reviews document.

---

## [Decision Letter · Decision Letter 1]

26 Nov 2025

Dear Dr. Steele,

Thank you for submitting a revised version of your manuscript to PLOS ONE. One of the original reviewers has evaluated the new version. Hei is generally happy with it, but has a few comments. Therefore, we invite you to submit a revised version of the manuscript that addresses these points.

We look forward to receiving your revised manuscript.

Kind regards,

Geir Ottersen

Academic Editor

PLOS ONE

Journal Requirements:

Reviewers' comments:

Reviewer's Responses to Questions

**Comments to the Author**

Reviewer #1: All comments have been addressed

2. Is the manuscript technically sound, and do the data support the conclusions?

Reviewer #1: Yes

3. Has the statistical analysis been performed appropriately and rigorously?

Reviewer #1: Yes

4. Have the authors made all data underlying the findings in their manuscript fully available?

Reviewer #1: Yes

5. Is the manuscript presented in an intelligible fashion and written in standard English?

Reviewer #1: Yes

Reviewer #1: Review of revised version of “"Identifying and assessing nonlinear drivers of capelin and Atlantic cod population dynamics using Empirical Dynamic Modelling (EDM) scenario exploration" by Steele et al.

This version is a great improvement over the previous version I reviewed. Most of the comments have been addressed adequately. The authors have clarified that my main concern regarding over-parametrization turned out to be a misunderstanding. I would generally advise on being more explicit about this in the manuscript. However, this clarification also raised another concern regarding multiple testing: a range of covariates were tested. This leads to increased probability of false positives, which I think can be addressed by some additional discussion (around line 245).

**Do you want your identity to be public for this peer review?** For information about this choice, including consent withdrawal, please see our Privacy Policy

Reviewer #1: No

You may also use PLOS’s free figure tool, NAAS, to help you prepare publication quality figures: https://journals.plos.org/plosone/s/figures#loc-tools-for-figure-preparation

---

## [Author Response · Author response to Decision Letter 2]

4 Dec 2025

See attached response to reviews file

---

## [Editor Report · Decision Letter 2]

7 Dec 2025

Identifying and assessing nonlinear drivers of capelin and Atlantic cod population dynamics using Empirical Dynamic Modelling (EDM) scenario exploration

PONE-D-25-37653R2

Dear Dr. Steele,

We’re pleased to inform you that your manuscript has been judged scientifically suitable for publication and will be formally accepted for publication once it meets all outstanding technical requirements.

Kind regards,

Geir Ottersen

Academic Editor

PLOS One

---

## [Editor Report · Acceptance letter]

PONE-D-25-37653R2

PLOS One

Dear Dr. Steele,

I'm pleased to inform you that your manuscript has been deemed suitable for publication in PLOS One. Congratulations! Your manuscript is now being handed over to our production team.

Kind regards,

on behalf of

Dr. Geir Ottersen

Academic Editor

PLOS One